# Effect of Different Doses of Vitamin D on the Intestinal Flora of Babies with Eczema: An Experimental Study

**DOI:** 10.3390/life12091409

**Published:** 2022-09-09

**Authors:** Youping Liu, Meng Yang, Zhiling Ran, Junxia Wang, Wujie Ma, Qiaoni Sheng

**Affiliations:** 1Department of Biochemistry and Molecular Biology, School of Basic Medical Sciences, Southwest Medical University, Luzhou 646000, China; 2Department of Pediatrics, The Affiliated TCM Hospital, Southwest Medical University, Luzhou 646000, China

**Keywords:** vitamin D, infantile eczema, intestinal flora, immune function

## Abstract

Infantile eczema is a common allergic disease caused by a variety of factors, which is often accompanied by immune dysfunction and dysbiosis of the intestinal flora. Vitamin D may affect the composition and function of intestinal flora by regulating the expression of antimicrobial peptides, thereby avoiding intestinal dysbiosis. The present study aims to explore whether the disorder of intestinal flora and immune function can be reversed by changing the Vit D intake of eczema infants. In this study, 12 healthy infants were selected as the healthy control group (CON), and 32 infants with eczema were selected for the eczema patient groups, of which 8 were randomly allocated as the eczema model group (ECZ, for which the infants’ peripheral blood and stool were collected before any treatment). The 12 healthy infants and 32 eczema infants all regularly adhered to the feeding of Vit D 400 IU/d. The 32 eczema infants were randomly divided into 3 groups, and patients in each group took Vit D 200 (D-LOW), 400 (D-MED), and 800 (D-HIGH) IU/day for 1 month, respectively. The peripheral blood and stool of the three groups were collected one month later. Flow cytometry was used to detect the levels of T lymphocyte subsets (CD4+, CD8+, and CD4+/CD8+) and serum inflammatory factor interleukin IL-6, IL-10, and interferon-γ(IFN-γ). The contents of serum immunoglobulin Ig E and 25-(OH) D_3_ were detected by chemiluminescence. Two hypervariable regions of the bacterial 16S rRNA gene (V3–V4) were high-throughput sequenced for stool intestinal flora analysis. The results showed that no significant difference was found in the content of 25 (OH) D_3_ between the ECZ and the CON groups. However, the intestinal flora and immune function in the ECZ group were remarkably more disordered than those in the CON group (*p* < 0.05). After the corresponding medical treatments for one month, the LOW-D and HIGH-D groups presented some reversals in the intestinal flora and immune-related indexes in comparison to the ECZ group, and the reversal effect in the LOW-D group was most significant (*p* < 0.05). These results indicated that low-dose Vit D(200 IU/d) can partly improve the disorder of intestinal flora and immune function in eczema infants who usually adhere to a Vit D preventive dose of 400 IU/d feeding.

## 1. Introduction

Intestinal flora is a large number of microorganisms living in the intestinal tract of the body, and its number is 10-fold the number of human cells [1] and about 100-fold the number of human genes [2]. Such a large number of intestinal flora and host form a “super organism”, that is, a complex interdependence is formed between the mammalian host and symbiotic intestinal microorganisms through interaction to achieve ecological sharing. The normal intestinal flora can be mainly classified into four phyla, i.e., firmicutes, bacteroidetes, proteobacteria, and actinobacteria. The four phyla account for 98% of the total intestinal flora [2]. Research has found that the human intestinal flora includes several hundred species of bacteria, but only just over 100 species of bacteria have been identified. According to their functions, they can be roughly divided into three categories, as follows: beneficial bacteria (such as *Bifidobacterium*, *Lactobacillus*, *Lactococcus*, *Enterococcus Faecium*, *Enterococcus fecalis*, *Eubacterium*, *Bacteroides vulgatus*, *Peptococcus*, etc.), harmful bacteria (such as *Escherichia coli*, *Staphylococcus*, *Prevotella*, *Proteus*, *L. welshimer*, *Verllonella*, *Clostridium Prazmowski*, etc.), and conditional pathogenic bacteria (such as *Escherichia coli*, *Enterococcus*, *Rumenococcus, Bacteroides*, *Desulfovibrio*, *Monilia albican*, etc.) [3]. Intestinal flora has increasingly been recognized as a key factor affecting human health [4]. Studies have shown that having disordered intestinal flora increases the likelihood of getting a disease [1,5].

Eczema is a common allergic disease caused by a variety of internal and external factors. The incidence rate of eczema in infants is high, especially in infancy [6]. Although the pathogenesis of infantile eczema has not been fully elucidated, many reports have indicated that the occurrence of infantile eczema is often related to the immature development of the immune system, the imperfect development of the intestinal barrier, family genetics, environmental factors (drugs, smoking environment), and the baby’s diet [7]. Recent studies have suggested that the development of the gut microbiota is decisive to the proper function of the guts and the development of the immune system [8]. Therefore, the intestinal flora affects the occurrence and development of infantile eczema, and the treatment of infantile eczema is often achieved by regulating infants’ intestinal flora [9].

Vitamin D (Vit D) is a fat-soluble vitamin which needs to be further hydroxylated after being absorbed by the digestive tract to form the active form of 1,25-dihydroxyvitamin D_3_ [1,25 (OH)_2_D_3_]. Active Vit D not only plays an important role in promoting calcium and phosphorus absorption and bone growth and development, but is also a key regulator of innate immune response to microbial threats [10]. In addition, it was found that Vit D can affect the composition and function of intestinal flora by regulating the expression of antimicrobial peptides, thus, avoiding intestinal flora imbalance [11]. Due to the low content of Vit D in breast milk, it cannot meet the needs of rapid growth and development of infants and young children. Therefore, the baby is routinely given a certain dose of Vit D (preventive dose 400 IU/d) until the age of two years [12]. Infantile eczema is often accompanied by immune dysfunction and dysbiosis of the intestinal florae [7]. Therefore, this study aimed to investigate if changing the dose of vitamin D can make the intestinal flora and immune function of eczema infants who usually adhere to Vit D 400 IU/d feeding return to normal levels. The results may provide a clinical reference and theoretical basis for infants with eczema on how to use Vit D more reasonably.

## 2. Materials and Methods

**Experimental drugs** were as follows: Oral medicine—vitamin D_3_ drops (capsule type) were purchased from Qingdao Shuangjing Pharmaceutical Co., Ltd., with the specification of 400 IU/capsule, 30 capsules/box, and the approval number was Guoyao Zhunzi H20113033. Bingbai liquid, a traditional Chinese medicine for external use, is a self-made external drug in the Hospital of Traditional Chinese Medicine affiliated to the Southwest Medical University. The specification was 100 mL/bottle, and the approval number was Chuanyao Zhizi Z20080317.

**Sampling** was carried out as follows: The present study applied convenience sampling and recruited 32 infants with eczema treated in the outpatient department of the affiliated Hospital of Traditional Chinese Medicine of Southwest Medical University from January 2021 to January 2022 as the participants. The recruitment adhered to the following strict inclusion and exclusion criteria:

Inclusion criteria were as follows: ① Patients who are full-term infants aged 1 to 12 months, receive exclusive breastfeeding, and adhere to the preventive dose of Vit D 400 IU/d; ② those who meet the diagnostic criteria of Western medicine [13], namely that the skin lesions are clustered or sporadic red papules, erythema, mound herpes, and other multiple lesions, which can be accompanied by small blisters, scales, crusts, exudation and skin pruritus, which occur repeatedly, and the acute and chronic phases overlap and alternate; ③ patients who are willing to accept the treatment plan and can actively cooperate with the follow-up; ④ patients whose legal guardian agreed and signed the informed consent form. 

The exclusion criteria were as follows: ① Patients who are participating in clinical trials of other drugs; ② patients who have been allergic to the test drug or have had serious allergic diseases in the past; ③ patients with poor compliance and unwilling to cooperate with treatment and follow-up; ④ patients with primary or congenital diseases, such as in the heart, brain, liver, and lung; ⑤ infants with eczema secondary to severe infection or infants with aggravation during treatment; ⑥ cases where there are factors which can lead to changes in the situation of infants, such as an unstable living environment, which is likely to cause loss of follow-up. 

At the same time, 12 healthy infants of the same age were selected as normal control subjects in the child protection department of the Affiliated Hospital of Traditional Chinese Medicine of the Southwest Medical University (inclusion criteria were as follows: ① Infants aged ≥ 1 month and ≤12 months, who usually adhere to the preventive dose of Vit D 400 IU/d; ② those without congenital diseases or complicated with serious heart, lung, kidney, and other skin diseases; ③ those who are willing to accept the treatment plan and actively cooperate with the follow-up; ④ the legal guardian of the patient agrees and signs the informed consent). The research program was approved by Ethics Committee of the Affiliated Hospital of Traditional Chinese Medicine of the Southwest Medical University and conformed to the Helsinki Declaration. 

**During the grouping and intervention study**, 8 of the 32 infants with eczema (stool and blood were collected before any treatment) were randomly selected as the eczema model group (ECZ group). Then, 32 infants with eczema were randomly divided into three groups, as follows: the Vit D low-dose group (LOW-D group, who took half of the preventive dose for one month), the Vit D medium-dose group (MID-D group, who kept the preventive dose for one month), and the Vit D high-dose group (HIGH-D group, who took double the preventive dose for one month). At the same time, 12 healthy infants were classified as the normal control group (CON group). The specific grouping and treatment of experimental subjects in each group were shown in Table 1. The general data (gender, average month age, etc.) showed no evident differences between groups (*p* > 0.05)

The process for the collection and detection of the peripheral blood was as follows: Collect 2 mL of fasting peripheral venous blood of all experimenters in the morning according to Table 1. After standing at room temperature for 10~20 min, take the serum and freeze it in a −20 °C refrigerator. Finally, all serums were sent to the Chengdu Gaoxin Daan medical laboratory Co., Ltd. for testing of immune related biochemical indexes. The levels of T lymphocyte subsets (CD4+, CD8+, and CD4+/CD8+) and serum inflammatory factors interleukin-6 (IL-6), interleukin-10 (IL-10), and interferon -γ(IFN-γ) were detected by flow cytometry; the active metabolite of Vit D is 1,25 (OH)_2_D_3_, but its half-life in blood is too short (4 h). Therefore, whether the body is deficient in Vit D is often judged by detecting the concentration of intermediate metabolite 25 (OH) D_3_ in the process of activation. The contents of serum immunoglobulin E (Ig E) and 25 (OH) D_3_ were detected by chemiluminescence.

The collection and detection of the stool was carried out as follows: According to Table 1, 1 mL of fresh stool (the first stool center in the morning) of all subjects were collected in a sterile stool collection tube and placed in an ice bath. All collection tubes had to be frozen and stored in a −80 °C refrigerator within one hour. After all stool samples were collected, they were sent to Beijing MyGenostics Inc for intestinal flora analysis. The 16S rRNA V3-V4 region of intestinal flora was used for high-throughput sequencing to build a database (filtering the sequencing data to remove low-quality reads and contaminated sequences, such as mitochondria/chloroplasts). The analysis of intestinal flora includes the following contents: ① Operational taxonomic units (OTUs) cluster analysis. To acquire information about the species and genus information in the sequencing results of each group, OTU clustering division is carried out on the sequence data after flattening all samples (all sequences are divided into many groups according to their similarities, and a small group is an OTU). In this study, all sequencing reads were analyzed by OTU clustering, species annotation, and abundance analyses using the Qiime2 software (V.2020.11. The source code is available at https://github.com/qiime2, accessed on 1 July 2022). ② α-diversity analysis. Alpha(α) diversity refers to the richness or diversity of bacterial communities in a specific region or ecosystem. Indexes, such as Chao1, ACE, Shannon, Simpson, etc., are commonly measured to analyze the complexity of the composition of intestinal flora in each group. A high index value suggests a more abundant and diverse composition of the corresponding sample flora. In this study, Chao1 and ACE indexes were used to characterize the richness of intestinal flora, and Shannon and Simpson indexes were used to characterize the diversity of intestinal flora. ③ β-diversity analysis. Beta (β) diversity refers to the difference in the composition of intestinal flora between different groups, including their similarities and differences. To evaluate the similarities among the groups, the present study applied the principal coordinate analysis (PCoA) based on Bray–Curtis dissimilarity by utilizing the Qiime2 software platform (Version 2020.11), which is based on the OTU counts. ④ Species difference and the indicator species analysis. Difference analysis of flora abundance is a very important step in the analysis of intestinal flora. With this method, the important taxa in the floristic differences in certain groups could be found. In order to find the microbiota with significant difference in species abundance, LEfSe analysis was used to analyze the difference of microbiota composition between groups in this study. ⑤ The analysis of metabolic function of intestinal flora. The role of intestinal microbiota in health and disease is largely attributed to the collective metabolic activities of microorganisms. Thus, in order to further explore the differences in intestinal microbiota among groups, the metabolic function of intestinal microbiota was predicted and analyzed by using the metabolic interaction network of the mouse and human gut microbiota NJC19.

**Statistical analysis** was carried out as follows: Bacterial community analyses based on 16S rRNA genes were performed using the Qiime2 software package(Version 2020.11) and the Greengenes 13_8 dataset as reference. The results of serum biochemical indexes and the difference analysis of intestinal flora were analyzed using SPSS 22.0 statistical software (IBM SPSS Inc., Chicago, IL, USA). The measurement data were expressed as mean ± standard deviation ($\bar{x}$ ± s). Data comparisons between multiple groups were analyzed by one-way ANOVA, and a SNK-q test was used for comparison between two groups. Here, *p* < 0.05 were considered statistically significant.

## 3. Results

### 3.1. The Relevant Biochemical Detection Indexes of Peripheral Serum in Each Group

The results of the biochemical detection indexes of peripheral serum in each group are shown in Table 2. The results of the CD4+/CD8+ ratio, IL-6, and Ig E in the ECZ group were comparatively higher than those in the CON group (*p* < 0.05). After being given different doses of Vit D for one month, the LOW-D, MID-D, and HIGH-D groups presented some degrees of reversal for all the above indexes in comparison to the ECZ group. The reversal effect of the LOW-D group was most significant (*p* < 0.05). The results of other test indexes have no significant difference among the five groups. 

### 3.2. The Results of Intestinal Flora Analysis in Each Group

#### 3.2.1. The Results of OTUs Cluster Analysis in Each Group

The results of the OTU cluster analysis in each group are shown in Figure 1. The results show that the dominant strains of infant intestinal flora at the phylum level are proteobacteria, actinobacteria, firmicutes, and bacteroides. These are also the four most representative phyla of human intestinal flora [2].

At the same time, the common proportion of OTUs in the CON group compared with the other four groups was preliminarily observed through inter-group cluster analysis. The results are shown in a Venn diagram in Figure 2. The results showed that, compared with the CON group, the ECZ group had a common proportion of 21.3% OTUs (Figure 2A), the LOW-D group had a common proportion of 25.9% OTUs (Figure 2B), the MID-D group had a common proportion of 21% OTUs (Figure 2C), and the HIGH-D group had a common proportion of 24.1% OTUs (Figure 2D). The common proportion of OTUs in the LOW-D group and CON group was the largest.

#### 3.2.2. The Alpha Diversity of the Bacterial Communities in Each Group

The alpha diversity metrics for the bacterial communities in each group are shown in Figure 3: there was no significant difference in Chao1 and ACE among the five groups (Figure 3A,B). This shows that there is no significant difference in the richness of intestinal flora among the five groups. Comparing the Shannon and Simpson values of the five groups, it is found that the Shannon and Simpson values of the HIGH-D group are all lower, which are significantly different from those of CON group (*p* < 0.01) (Figure 3C,D). There is no significant difference between the other groups. It shows that the diversity or evenness of intestinal flora in the HIGH-D group is significantly lower than that of the CON group.

#### 3.2.3. The Beta Diversity Analysis of the Bacterial Communities in Each Group 

The results of the principal coordinates analysis (PCoA) of five groups of intestinal flora based on Bray–Curtis are shown in Figure 4. The results showed that the composition of intestinal flora was the most similar between the ECZ group and the MID-D group, and that the composition of intestinal flora was the most similar between the LOW-D group and the CON group. There were significant differences in the composition of intestinal flora among other groups.

#### 3.2.4. Species Difference and Marker Species Analysis of Intestinal Flora in Each Group

The comparison of relative abundance of marker species based on LEfSe analysis in each group are shown in Figure 5. The results showed that, compared with the CON group, the s-bacteroides vulgatus, s-phascolactobacterium faecium, o-acidaminococcales, f-acidaminococcaceae, g-phascolactobacterium, f-sutterellaceae, g-parasutterella, and s-parasutterella incrementihominis in other four groups decreased significantly, especially in the ECZ group (*p* < 0.01 or 0.05). After being given different doses of Vit D for one month, the LOW-D and HIGH-D groups presented some degrees of reversal for all the above bacteria in comparison to the ECZ group. The reversal effect of the LOW-D group was most significant (*p* < 0.05) (Figure 5: 1, 2, 3, 4, 5, 7, 9, 10). On the contrary, f-prevotellaceae, g-prevotella, and s-prevotella copri increased significantly in the other four groups compared with the CON group, and the increase was the most obvious in the ECZ group (*p* < 0.05). After being given different doses of Vit D for one month, the LOW-D and HIGH-D groups presented some degrees of reversal for the three bacteria in comparison to the ECZ group. The reversal effect of the LOW-D group was most significant (*p* < 0.05) (Figure 5: 6, 8, 11). The above results show that the composition and species abundance of intestinal flora in the LOW-D group are closest to those in the CON group, which is consistent with the results of the cluster analysis of OTUs between groups and the β-diversity analysis (Figure 2 and Figure 4).

#### 3.2.5. The Analysis of Metabolic Function of Intestinal Flora in Each Group

The metabolic function analysis of the intestinal flora in each group is shown in Figure 6. The results showed that there were some regular differences in the metabolic function of intestinal flora among the five groups. Compared with the CON group, the D-glucose consumption, raffinose consumption, lactose consumption, maltose consumption, and succinate production of intestinal flora in the ECZ group increased significantly (*p* < 0.05). After being given different doses of Vit D for one month, the LOW-D and HIGH-D groups presented some degrees of reversal for all the above metabolic function in comparison to the ECZ group. The reversal effect of the LOW-D group was most significant (*p* < 0.05) (Figure 6: 1, 2, 3, 4, 6). The CO_2_ production in the ECZ group was lower than that of the CON group (*p* < 0.05) (Figure 6: 5). After being given different doses of Vit D for one month, the LOW-D group presented some degree of reversal for the above metabolic function in comparison to the ECZ group(*p* < 0.05) (Figure 6: 5). In other words, the metabolic function analysis of intestinal flora shows that the LOW-D group is closest to the CON group, which is consistent with the results of the cluster analysis of OTUs between groups, the β-diversity analysis, and the analysis of relative abundance of marker species. (Figure 2, Figure 4 and Figure 5).

## 4. Discussion

This study found that there were significant differences in the composition and function of intestinal flora of infants in the ECZ group and CON group. In the ECZ group, various beneficial bacteria, such as s-bacteroides vulgatus [3,14,15], s-phascolactobacterium faecium [16], o-acidaminococcales [17], f-acidaminococcaceae [18], g-phascolactobacterium [19], f-sutterellaceae [20], g-parasutterella [21], and s-parasutterella incrementihominis [22] decreased significantly compared with the CON group (*p* < 0.01 or 0.05). Among them, bacteroides vulgatus and acidaminococcales can produce short chain fatty acids (SCFA, such as acetic, propionic, and butyric acids) together with various gases (carbon dioxide, methane, etc.) by fermenting non-digestible carbohydrates [15,17,23]. The SCFA produced by fermentation can provide energy for the host, maintain intestinal homeostasis, and participate in immune regulation, which is of great significance to human health [17,24]. In addition, bacteroides vulgatus is one of the main bacteria that use dietary fiber fermentation to release more carbon dioxide [14,25]. Therefore, in the metabolic function analysis of intestinal flora, it is found that the significant reduction in carbon dioxide production in the ECZ group (Figure 6: 5) may be related to the significant reduction in Bacteroides vulgatus in the ECZ group. Additionally, F-sutterellaceae [20], g-phascolactobacterium [19], and s-phascolactobacterium faecium [16] were negatively correlated with inflammatory response. Here, g-parasutterella is defined as the core component of healthy fecal microbiota in the human gastrointestinal tract, which contains the following two types of strains: parasutterella incrementihominis and parasutterella secunda [22]. It was found that intestinal colonization of parasartrella was positively correlated with intestinal bile acid metabolites and aromatic amino acid metabolites. Intestinal bile acid metabolites were involved in host lipid metabolism and the digestion and absorption of fat soluble vitamins [21]. Aromatic amino acid metabolites could enhance the integrity of intestinal epithelial barrier [21].It can be seen that the most significant reduction in the above various beneficial bacteria occurred in the ECZ group (Figure 5: 1, 2, 3, 4, 5, 7, 9, 10) (*p* < 0.01 or 0.05), which will lead to a sharp reduction in intestinal microbial metabolites beneficial to the body, such as SCFA, aromatic amino, acids and bile acids in the intestine of infants in ECZ group.

Compared with the CON group, the levels of f-prevotellaceae, g-prevotella, and s-prevotella copri in the ECZ group were significantly higher (Figure 5: 6, 8, 11) (*p* < 0.05). They belong to the same family of prevotella. Chen Junkui found that prevotella is highly dependent on a medium rich in glucose, galactooligosaccharide, maltooligosaccharide, and raffinose [26], and that prevotella series bacteria can produce a large amount of succinic acid by fermenting the above carbohydrates [27]. It can be seen that the increased consumption of glucose, raffinose, lactose, and maltose, and the increased succinic acid production in the ECZ group (Figure 6: 1, 2, 3, 4, 6) (*p* < 0.05) may be related to the significant increase in prevotella series bacteria in the ECZ group. These elevated prevotella are all harmful bacteria [3]. Some studies have found that prevotella is negatively correlated with host health [28] and positively correlated with inflammatory diseases [29]. They can induce disease and inflammation by increasing the release of immune cells and various inflammatory mediators and, thus, prevotella is also known as a proinflammatory bacteria [30]. It can be seen that the significant increase in prevotella in the ECZ group will lead to the release of various inflammatory mediators in infants. At the same time, the significant decline of important beneficial bacteria, such as bacteroides, vulgaris, and parasartrella in the ECZ group led to the sharp decline of beneficial metabolites, such as SCFA, aromatic amino acids, and bile acids. The previously mentioned disorder of intestinal flora may be the mechanism triggering the occurrence and development of eczema in infants who are fed with the preventive dose of Vit D (400 IU/d).

Vitamin D can affect the composition and function of intestinal flora [11]. Therefore, we expect to reverse the disorder of intestinal flora in eczema infants by changing the feeding dose of Vit D. This study found that after being given different dosages of Vit D for one month, the LOW-D and HIGH-D groups presented some reversals of the intestinal flora disorder in comparison to the ECZ group, and the reversal effect in the LOW-D group was most significant (*p* < 0.05) (Figure 2, Figure 4, Figure 5 and Figure 6).

The concentration of 25 (OH) D_3_ in serum is used clinically as an indicator of Vitamin D level. The thresholds for 25 (OH) D_3_ serum levels (<20 ng/mL for severe deficiency, 20~30 ng/mL for deficiency, and >30 ng/mL for sufficient amounts) were derived from children [31]. This study found that the serum level of 25 (OH) D_3_ in the ECZ group was basically >30 ng/mL, and there was no significant difference compared with the CON group (as shown in Table 2). It indicates that Vit D in eczema infants is sufficient, which may be related to the usual preventive dose of Vit D supplementation (400 IU/d). At the same time, the content of active Vit D [1,25 (OH)_2_D_3_] in the body is strictly monitored. That is, a regulatory loop formed by the coordination of parathyroid hormone (PTH), fibroblast growth factor 23 (FGF 23), and vitamin D receptor (VDR), so as to ensure that the synthesis amount of active Vit D can be stabilized in a reasonable range, and over production or insufficient supply of active Vit D is avoided [32]. It can be seen that eczema infants fed with the preventive dose of Vit D (400 IU/d) are generally not prone to Vit D deficiency.

It is speculated that the occurrence of infantile eczema (who always adhere to Vit D 400 IU/d feeding) is not caused by Vit D deficiency, but likely by the disorder of intestinal flora. Because the intestinal flora in infants is very unstable and undergoes continuous succession after birth, it is easy for the structure of intestinal flora to become disordered and break the balance of the immune system, resulting in the occurrence and development of allergic diseases, such as eczema [33]. This study found that eczema infants not only had significant intestinal flora disorders, but that they also had certain immune system disorders. The serum immune function related indexes, namely the CD4+/CD8+ ratio, Ig E level, and serum inflammatory factor IL-6, in the ECZ group were higher than those in the CON group (*p* < 0.05) (Table 2).After being given different doses of Vit D for one month, it was found that, for eczema infants with intestinal flora disorder (those who always adhere to Vit D 400 IU/d feeding), increasing the feeding dose of Vit D in the HIGH-D group (800 IU/d) still could not significantly reverse the disorder of immune function (Table 2) and intestinal flora (Figure 2, Figure 4, Figure 5 and Figure 6). This further verifies that the disorder of intestinal flora in infants with eczema (those who always adhere to Vit D 400 IU/d feeding) is not caused by Vit D deficiency. On the contrary, compared with the MID-D and HIGH-D groups, the LOW-D group (200 IU/d) had the best reversal effect of the disorder of immune function and intestinal flora (*p* < 0.05) (Table 2 and Figure 2, Figure 4, Figure 5 and Figure 6).

Vitamin D is a fat-soluble vitamin. Under normal circumstances, the intestine absorbs Vit D through the formation of bile acid-dependent particles [34]. This study found that the parasartrella, which is positively related to intestinal bile acid metabolites and aromatic amino acid metabolites, decreased greatly, which will directly result in the blocking of the transfer of intestinal Vit D to the gastrointestinal mucosa, and, thus, the limiting of the absorption of vitamin D in the first stage [35,36]. At this time, it is useless to feed more Vit D to infants with eczema. It further proves that it is of little significance to increase the feeding dose of Vit D (800 IU/d) for infants with eczema who have suffered from intestinal flora disorder (those who always adhere to Vit D 400 IU/d feeding). At this time, feeding infants with a low dose (200 IU/d) may reduce the gastrointestinal burden of infants with eczema.

Moreover, there is a bidirectional regulation between intestinal flora and the Vit D/VDR pathway. That is, intestinal flora can affect the Vit D/VDR signal pathway by regulating Vit D absorption, hydroxylation, active Vit D degradation, and VDR expression [37]. Conversely, the Vit D/VDR signaling pathway can also affect the composition and function of intestinal flora [11,38]. It was found that in immune cells, such as macrophages, CYP27B1 (25 hydroxyvitamin D 1α-Hydroxylase) expression is induced by immune specific input, which leads to the local production of 1,25 (OH)_2_D_3_ at the infected site, and directly induces the expression of genes encoding antimicrobial peptides [10]. The expressed antimicrobial peptides can participate in regulating the composition and function of intestinal flora to avoid the imbalance of intestinal flora [11]. Therefore, it is speculated that, for eczema infants with intestinal flora disorder (who always adhere to Vit D 400 IU/d feeding), changing their Vit D to low-dose (200 IU/d) feeding may reduce the gastrointestinal burden of infants, and improve the composition of intestinal flora by improving the Vit D/VDR signal pathway. This will improve the immune function and eczema symptoms of infants by increasing the abundance of beneficial bacteria, such as Bacteroides vulgatus, and by decreasing the abundance of the proinflammatory bacteria prevotella. Compared with the MID-D and HIGH-D groups, the LOW-D group (200 IU/d) had the best reversal effect of the level of the serum immune function related indexes CD4+/CD8+ ratio, Ig E, and serum inflammatory factor IL-6 (Table 2). In other words, the intestinal flora and immune related indexes of the LOW-D group and the CON group were the closest. These findings are also supported by the cluster analysis of OTUs between the groups, the β-diversity analysis, and the metabolic function of intestinal flora (Figure 2, Figure 4 and Figure 5).

The current study also has several limitations. As some patients’ legal guardians were unwilling to cooperate, the sample size was a little small. Furthermore, a clear definition of what would be the composition of a healthy microbiota is also required. Therefore, the impact of the gut microbiota on overall human health remains controversial [39]. Future studies may study need to expand the sample size and further explore the mechanism in detail.

## 5. Conclusions

In conclusion, the intestinal flora and immune function in infants with eczema (who usually adhere to the feeding of Vit D 400 IU/d) were remarkably more disordered than those in healthy infants (who usually adhere to the feeding of Vit D 400 IU/d), and the low-dose vitamin D (200 IU/d) contributes to the reversal of the disordered immune function and intestinal flora in eczema infants who adhere to a Vit D prophylactic dose of 400 IU/d feeding. This finding is worthy of further in-depth study.

## Figures and Tables

**Figure 1 life-12-01409-f001:**
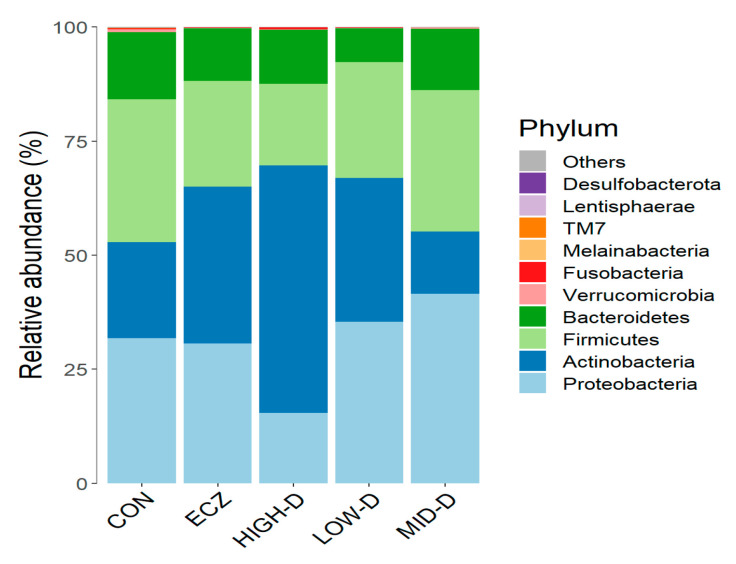
Relative abundance of microbiota in the top 10 at the phylum level in each group.

**Figure 2 life-12-01409-f002:**
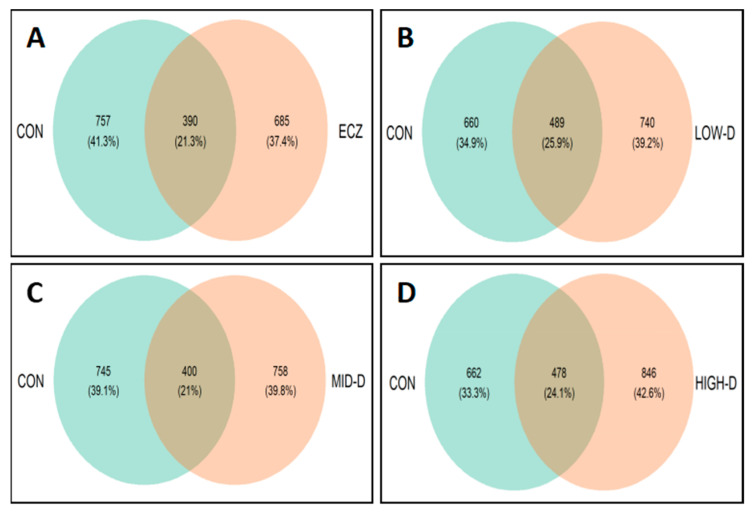
OTUs shared between CON and the other four groups in infant intestinal flora (**A**) ECZ vs. CON; (**B**) LOW-D vs. CON; (**C**) MID-D vs. CON; (**D**) HIGH-D vs. CON.

**Figure 3 life-12-01409-f003:**
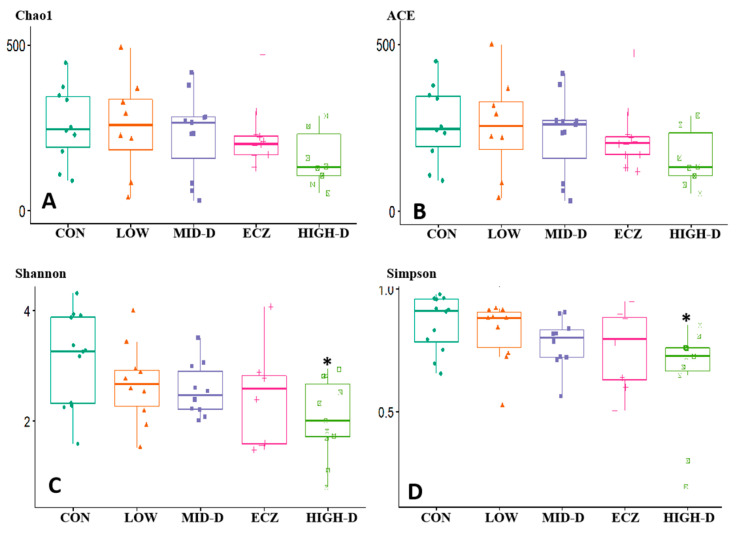
α-diversity analysis of intestinal flora in each group. (**A**) Chao1; (**B**) ACE; (**C**) Shannon; (**D**) Simpson. The Chao1 and ACE characterized the abundance of intestinal flora; Shannon and Simpson characterized the diversity of intestinal flora. * *p* < 0.01 (HIGH-D vs. CON)].

**Figure 4 life-12-01409-f004:**
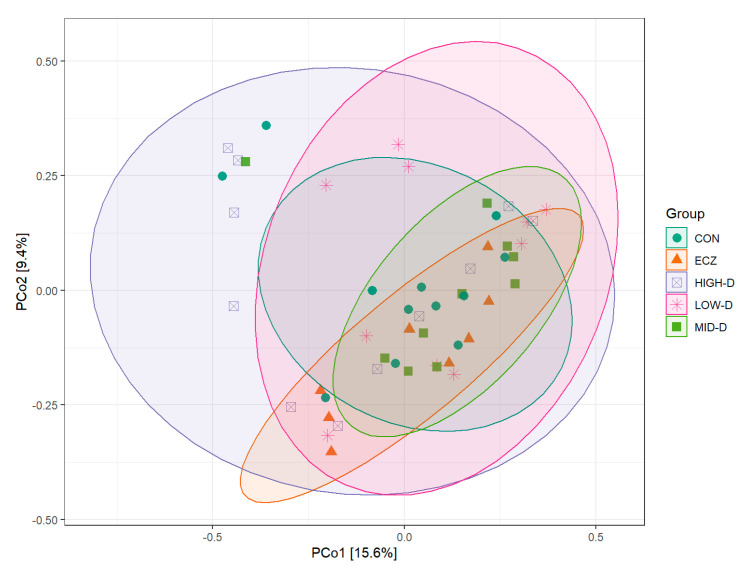
β-diversity analysis of intestinal flora in each group based on PCoA. The figure shows the two-dimensional diagram of PCoA, in which each dot represents a sample, and each colored geometric shape represents a group. Here, PCo1 is the principal coordinate component causing the largest difference in samples, with an explanatory value of 15.6%, while PCo2 comes second, with an explanatory value of 9.4%.

**Figure 5 life-12-01409-f005:**
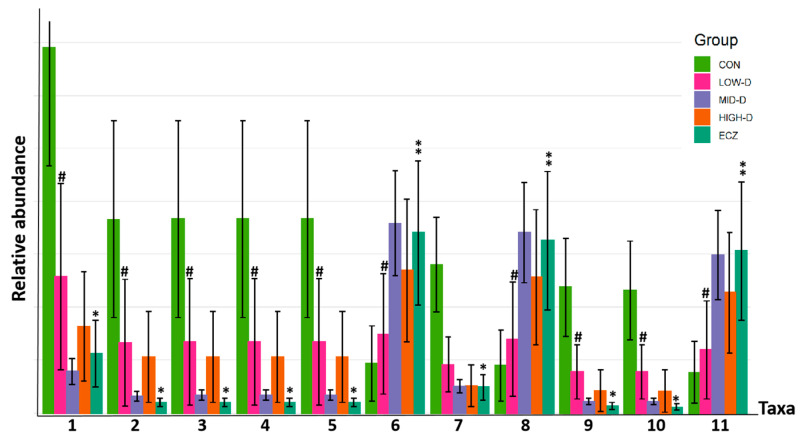
Comparison of relative abundance of marker species based on LefSe analysis in each group. (1) s-Bacteroides vulgatus; (2) s-Phascolarctobacterium faecium; (3) o-Acidaminococcales; (4) f-Acidaminococcaceae; (5) g-Phascolarctobacterium; (6) f-Prevotellaceae; (7) f-Sutterellaceae; (8) g-Prevotella; (9) g-Parasutterella; (10) s-Parasutterella excrementihominis; (11) s-Prevotella copri; * *p* < 0.01 (ECZ vs. CON); ** *p* < 0.05 (ECZ vs. CON); ^#^
*p* < 0.05 (LOW-D vs. ECZ).

**Figure 6 life-12-01409-f006:**
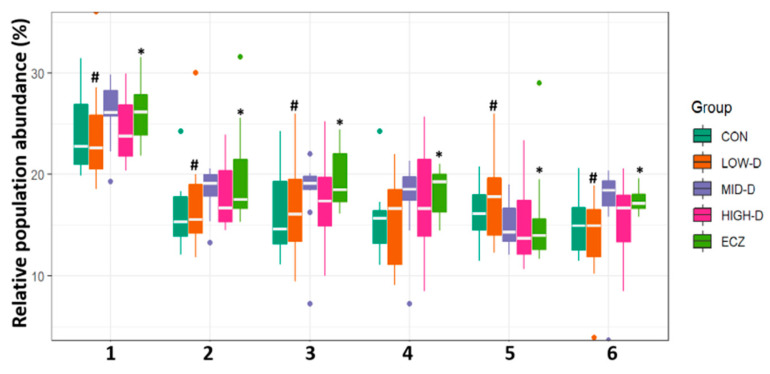
Analysis on the difference of metabolic function of intestinal flora in each group. (1) D-Glucose consumption; (2) Raffinose consumption; (3) Lactose consumption; (4) Succinate production; (5) CO2 production; (6) Maltose consumption. * *p* < 0.01(ECZ vs. CON); ^#^
*p* < 0.05(LOW-D vs. ECZ).

**Table 1 life-12-01409-t001:** Research object grouping and corresponding treatment.

Group	N(M/F)	Average Month Age (Months)	Stool and Peripheral Blood Were Collected Before any Treatment	Application of Traditional Chinese Medicine Bingbai Liquid on the Affected Area	Oral Vit D (400 IU/capsule) for One Month	One Month Later, Peripheral Blood and Stool Were Collected
CON	12(4/8)	5 ± 1.21	+	-	-	-
ECZ	8(3/5)	5 ± 1.53	+	-	-	-
LOW-D	11(4/7)	6 ± 1.13	-	+	200 IU/d (One capsule for two days)	+
MID-D	10(3/7)	5 ± 0.78	-	+	400 IU/d (One capsule a day)	+
HIGH-D	11(3/8)	6 ± 1.44	-	+	800 IU/d (Two capsules a day)	+

Note: The purpose of applying Bingbai liquid is only to treat symptoms, such as dry skin, itching, etc. Here, M/F indicates male/female).

**Table 2 life-12-01409-t002:** Comparison of biochemical detection indexes in each group ($\bar{x}$ ± s).

Group	n	CD4+(%) (27–51)	CD8+(%) (15–44)	CD4+/CD8+(%) (0.71–2.78)	Ig E(IU/mL) (<29)	IL-6(pg/mL) (<18.6)	IL-10(pg/mL) (<12.6)	INF-γ(ng/L) (184.9–909.8)	25 (OH) D_3_ (ng/mL) (>20)
CON	12	39.87 ± 1.82	20.23 ± 1.53	1.77 ± 0.21	25.4 ± 2.52	10.37 ± 0.03	6.04 ± 0.63	502.64 ± 23.66	39.86 ± 3.67
ECZ	8	46.46 ± 2.61	15.99 ± 1.02	3.64 ± 1.26 *	48.0 ± 6.37 *	21.09 ± 1.85 *	8.26 ± 1.12	639.98 ± 35.61	31.86 ± 3.11
LOW-D	11	40.95 ± 2.12	18.60 ± 1.32	2.11 ± 0.39 ^#^	26.9 ± 2.55 ^#^	12.19 ± 1.23 ^#^	5.95 ± 1.33	611.23 ± 18.34	32.57 ± 4.02
MID-D	10	45.22 ± 3.61	16.54 ± 1.47	3.16 ± 0.41	35.4 ± 3.89	17.16 ± 1.63	8.45 ± 1.89	641.21 ± 26.63	34.04 ± 3.43
HIGH-D	11	41.18 ± 2.73	17.61 ± 1.63	3.32 ± 0.67	38.0 ± 3.11	17.61 ± 1.77	7.04 ± 1.53	702.64 ± 31.12	41.96 ± 4.15

Abbreviations are as follows: CD (cluster of differentiation), Ig E (Immunoglobulin E), IL-6 (Interleukin-6), IL-10 (Interleukin-10), INF-γ (Interferon-γ). * *p* < 0.05 vs. CON; ^#^
*p* < 0.05 vs. ECZ.

## Data Availability

The datasets presented in this study can be found in online repositories (Appendix A). The names of the repository/repositories and accession number(s) can be found below: NCBI database, accession number PRJNA848626.

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
