# Peer review of "Effect of Different Doses of Vitamin D on the Intestinal Flora of Babies with Eczema: An Experimental Study"

_life, 2022, doi:10.3390/life12091409_

Round 1

Reviewer 1 Report

Liu et Al. performed a study to explore whether the disorder of intestinal florae and immune function can be reversed by changing the Vit D intake of eczema infants. To my opinion the topic is very in depth studied, with a robust statistical analysis and well discussed. I have some suggestion before the publication.

1)    To my opinion an important limitation of the study is the missing of baseline clinical characteristics (i.e. maternal supplementation of Vit D, term of preterm newborn, breastmilk or formula). It should be added in limitation section and discussed. In conclusion section caution should be used, the other risk of vit D supplementation are missing (i.e. growth parameters, infection disease). “If it is not caused by Vit D deficiency, it is recommended to feed low-dose Vit D (200 IU/d) for a period of time”, to my opinion is too strong or should be more justified or discussed;

2)    Despite there is a lot of figures in the manuscript, maybe it could be nice transferred data of tables 2 and 3 in two figures;

3)    There are sentences with references missing  lines 37-40; 49-51; 305-307; 336-338; 

4)    Typing error: line 17 “?” and line 350 “- -"

Hong et al. Infenction 2022 (https://doi.org/10.1007/s15010-022-01845-4)

Abramsn Early Hum Deb 2021 (https://doi.org/10.1016/j.earlhumdev.2021.105461)

Kumar Pediatrics 2022 (https://doi.org/10.1542/peds.2022-057092k)

Author Response

请参阅附件。

Reviewer 2 Report

In this study, Youping Liu, et al. observed the changes of intestinal flora and immune function after feeding different doses of Vit D to infants with eczema, so as to provide clinical reference and theoretical basis for infants with eczema on how to use Vit D more reasonably. They conclude that low-dose Vit D (200IU/d) can improve the disorder of intestinal flora and immune function in eczema infants to a certain extent.

Strengths of the study:

-       Study question is valid

-       Adequate literature review was performed.

-       The research results validate the author's conclusions.

The manuscript can be improved by addressing following concerns.

-       My only suggestion is that authors may want to provide the baseline clinical characteristics of infants (like duration of symptoms, distribution, severity, medications used, any other allergies present, etc) in the eczema model group and vitamin D groups (Low-D, Mid-D and High-D), and if there is any significant difference in the characteristics. Study results may not be valid if patient populations are not comparable. 

Reviewer 3 Report

Even though the idea sounds interesting, some points need clarification, refinement, reanalysis, rewriting and more information to improve this article.

 Major points

The manuscript needs writing and language editing. The title should improve, for example, “Effect of different doses of vitamin D on the intestinal flora of babies with eczema: A experimental study”. The main objective must be the same throughout the manuscript (abstract, introduction and results/discussion). The abstract should summarize the main points of this experimental study. Authors should not use the words that appear in the title as keywords. References must be recent and relevant.

1.    The introduction section is too concise. The authors need to delve into the topics covered (eczema, microbiota, and vitamin D intake). What are the most representative phyla of the human intestinal flora, in healthy infants? Is there any information on the microbiota in patients with eczema or other eczema-related allergic diseases? What is the difference between alpha and beta diversity? Is there any other diversity to take into account? The research question should be clearly outlined. A good and clear justification for conducting this study should be given. It would be better if the authors offered a hypothesis before the main objective of this study.

2.   The materials and methods section needs improvement. More details are required to replicate this study. What type of study was it? Was it an experimental study, right? Why was this experiment not a double-blind study? It would be a good idea to use more subsections, such as Population Study, Ethical Considerations, Sampling, Assessment Study, Intervention Study, Laboratory Procedures, Statistical Analysis, etc. It would be better to show the inclusion and exclusion criteria for the patient group and the control group in separate paragraphs. Why were these patients (heart, brain, liver, and lung) excluded? give reasons. How did the authors determine the number of patients (total/per group) for there to be statistical significance? In the experimental phase, patients and controls were classified into five groups as follows:… Table 1 shows the analyzes and treatments performed in each study group. In Table 1: The application of Bingbai liquid of traditional Chinese medicine in the affected area was used simultaneously with the treatment of vitamin D? The authors assumed that patients with eczema would achieve similar values in peripheral blood and faeces as healthy patients when treated with vitamin D? 400UI (One capsule every other day). How many phyla did the authors consider could be submitted/studied? What are the normal cutoff points for blood variables? Specify the brand, city, and country of the software. Was a record of patient and control data performed? Was there any kind of clinical evaluation of the patients and controls before and after the experimental study? Why didn't the authors study the diet of the patients, if the diet is an important factor that contributes to maintaining/regulating the microbiota? What was the washout period prior to the Vit D dose change? one month? The authors should specify that in this experimental study, to show the effect of different doses of Vit D on the intestinal flora of eczema patients, a washout period was considered to avoid the bias of the previous doses of Vit D taken by the patients. Also, specify if the doses were taken daily or once a week, etc. The statistical analysis needs improving.

3.    In the results section: In the text, the authors should write the most significant results, and they should avoid repeating the same information in the text if this data appear in the tables, figures, etc. Lines 158-160: this description should appear in the M&M section. All the steps the authors took to obtain the results should be described in the M&M section, not here. The figures and tables must be of quality to see the names, and numbers, and the authors must write the full name of the abbreviations used in a legend at the bottom of the figures. It would be better to show in Figure 2, in each group, the common proportion (number) between them and the control group. Avoid repetitions. This common proportion was significantly different from other groups. Lines 181-185, 200-202, 211-213 and 243-244: it would be better to write this information in the introduction section. Lines 185-187, 202-203, 213-215 and 243-246: it would be better to write this information in the M&M section. All the appreciations that the authors make of the results are written in the discussion section. Lines 248-250: What material/method did the authors use to measure these variables/outcomes (write them in M&M)?

4.   The discussion needs deep improvements, and the discussion must be more argumentative in relation to the main goal of this study. This section should start with the main objective of this study and the most significant results. The results must be discussed from multiple angles and placed in context without being over-interpreted. Lines 273-275: It would be better to write this information in the M&M. The discussion should be shortened based on the objectives and the most significant results that lead us to the same conclusion reached by the authors. A paragraph of limitations and suggestions should be written before the conclusions.

5.   The conclusion needs to improve.

Minor points are highlighted in the accompanying manuscript.

 I encourage the authors to rewrite the manuscript, thinking about the principal goal of this study, its design and answering with the results and arguments of the discussion the most proper conclusion to this research work.

Round 2

Reviewer 1 Report

.

Reviewer 3 Report

Even though the authors have made significant improvements, the article has not met the journal's recommended guidelines. For this reason, I encourage authors to consider the following suggestions aimed at improving the presentation of their experimental study and its subsequent publication.

Major points

The writing of the manuscript must improve. It must be concrete, concise, and detailed.

The abstract should summarize the main points of this experimental study, for example, 32 babies were classified into five groups: four eczema patient groups (ECZ) and one control group (CON). The patients per group took 200 (D-LOW), 400 (D-MED), and 800 (D-HIGH) IU/day for one month, respectively. What is the most significant result (conclusion)?

1.    The introduction section: It would be better if the authors offered a hypothesis before the main objective of this study: Can the intestinal flora and immune function of children with eczema return to normal levels by changing the dose of vitamin D? (Hypothesis)

2.   The materials and methods section: It would be a good idea to use more subsections, such as Population Study, Ethical Considerations, Sampling, Assessment Study, Intervention Study, Laboratory Procedures, Statistical Analysis, etc. The authors must specify that in this experimental study the collection of patients was through convenience sampling. Please write the inclusion and exclusion criteria in separate paragraphs. The doses administered to the experimental groups are not clear. Authors should describe those patients in the LOW-D group who took half the dose of Vit D, that those in the MED-D group who took the same dose, and those in the HIGH-D group who took twice the dose (preventive), a day for a month. Authors should add references to added information.

3.    In the results section: How many babies older than 6 months and boys were there? The serum level of 25 (OH) D3 in the ECZ group was basically > 30 ng/mL, and there was no significant difference compared with the CON group. The size of the figures could be increased to be able to see them better. PCoA: Authors should be aware that all procedures and analyses of the study population should be described in detail and referenced in the M&M section. The D-glucose-consumption, raffinose-consumption, lactose-consumption, maltose-consumption and succinate-production by groups: What material/method did the authors use to measure these variables/outcomes?  

4.   The discussion: this section should start with the main objective of this study and the most significant results. The results must be discussed from multiple angles and placed in context without being over-interpreted. The diet could be a bias.

5.   The conclusion should be improved.

Minor points are highlighted in the accompanying manuscript.
